# The impact of the land-to-sea transition on evolutionary integration and modularity of the pinniped backbone

Juan Miguel Esteban [1✉], Alberto Martín-Serra [1], Alejandro Pérez-Ramos [1], Baptiste Mulot[2], Katrina Jones[3] & Borja Figueirido [1]

In this study, we investigate how the terrestrial-aquatic transition influenced patterns of axial integration and modularity in response to the secondary adaptation to a marine lifestyle. We use 3D geometric morphometrics to quantify shape covariation among presacral vertebrae in pinnipeds (Carnivora; Pinnipedia) and to compare with patterns of axial integration and modularity in their close terrestrial relatives. Our results indicate that the vertebral column of pinnipeds has experienced a decrease in the strength of integration among all presacral vertebrae when compared to terrestrial carnivores (=fissipeds). However, separate integration analyses among the speciose Otariidae (i.e., sea lions and fur seals) and Phocidae (i.e., true seals) also suggests the presence of different axial organizations in these two groups of crown pinnipeds. While phocids present a set of integrated "thoracic" vertebrae, the presacral vertebrae of otariids are characterized by the absence of any set of vertebrae with high integration. We hypothesize that these differences could be linked to their specific modes of aquatic locomotion –i.e., pelvic vs pectoral oscillation. Our results provide evidence that the vertebral column of pinnipeds has been reorganized from the pattern observed in fissipeds but is more complex than a simple "homogenization" of the modular pattern of their close terrestrial relatives.

[1] Departamento de Ecología y Geología, Facultad de Ciencias, Universidad de Málaga, Campus de Teatinos s/n, 29071 Málaga, Spain. [2] ZooParc de Beauval & Beauval Nature, 41110 Saint-Aignan, France. [3] Department of Earth and Environmental Sciences, University of Manchester, Williamson Building, Oxford Road, Manchester M13 9PL, UK. ✉email: jmesteban@uma.es

The vertebral column is a semi-rigid structure of the axial skeleton that is formed by segmented series of bony elements (vertebrae) separated by mobile joints with intervertebral discs[1]. In land-going vertebrates, four distinct regions are usually recognized according to vertebral morphology and function (cervical, dorsal, sacral, and caudal)[2], but in therian mammals there are regional subdivisions within the trunk (dorsal series) into thoracic (=anterior dorsals) and lumbar (=posterior dorsals, in part) vertebrae[3]. In this regard, it has been hypothesized that the high regionalization of mammals evolved at the expense of count variability[3,4], with the cooption of existing underlying modules for new functions[5–7]. This cooption increases evolvability and complexity in these anatomically-specialized regions (e.g. refs. [4,8,9]). Accordingly, understanding the evolution of spinal regionalization and how regions get reorganized into underlying functional modules is key to investigate the evolution of mammalian gaits, their locomotor diversity, and their respiratory function (e.g. refs. [10–14]), among others biological issues (e.g. ref. [15]).

In this respect, recent analyses on the evolutionary integration and modularity of presacral vertebrae in terrestrial (fissipeds) carnivoran mammals have evidenced the presence of three underlying functional modules (cervical, anterodorsal, and posterodorsal)[16]. The last two modules likely relate to locomotor performance, as they appear to be related to motion capability of the presacral spine[16]. Particularly, the vertebrae of the anterodorsal module (i.e., thoracic vertebrae anterior to the diaphragmatic vertebra) may be related to motion constraints of the thorax, while the high integration observed for the posterodorsal vertebrae (i.e., those vertebrae posterior to the diaphragmatic vertebra) could be related to prevent the excessive extension that results from increasing vertebral motion at "Diaphragmatic joint complex" (a key region of the mammalian column of exceedingly permissive motion)[17–20]. Moreover, these authors also suggested that the diaphragmatic vertebra, which marks the limit between the anterodorsal and posterodorsal modules, was not integrated with any of the modules. Martín-Serra et al.[16] interpreted the lack of integration of the diaphragmatic vertebra as related to the motion ability of that region, named "Diaphragmatic joint complex"[18–20].

The land-to-sea transition is among the most extreme ecological shifts in mammalian evolution, and changes for enduring gravity and buoyancy are the most important physical aspects. Thus, aquatic tetrapods, while underwater, no longer support their body weight, nor do they have to locomote by generating friction with the substrate. Moreover, the vertebral column of secondarily aquatic tetrapods is more involved in locomotion and flexibility than that of fully-terrestrial tetrapods with appendicular locomotion[19–22]. Accordingly, the land-to-sea transition should have an impact on vertebral column integration and modularity. In the axial skeleton of some marine mammals, such as cetaceans, secondary aquatic adaptations involved the reduction of regionalization[23,24] in which lumbar, sacral, and anterior caudal vertebrae are integrated into a single "torso module" but also provided evidence of the conservation of regional identities (e.g., lumbar and caudal vertebrae can be visually discerned). Therefore, cetaceans not only experience de-differentiation (or homogenization) of their vertebral columns but also a clear reorganization of the existing modules into new ones.

In this sense, the evolution of the pinniped (Fig. 1) vertebral column is remarkable. While they present several adaptations for swimming and diving, pinnipeds still have some capacity to move on land, where they perform important activities, such as mating and giving birth[25]. Phocids swim using pelvic oscillation to generate thrust (hindlimb-dominated swimmers)[26–29] and they exhibit a terrestrial locomotion like the movement of

caterpillars[30]. Accordingly, the anterior region of the phocid column forms a rigid torso, and most of the intervertebral flexibility is restricted to the posterior region, which possess large epaxial muscles[31]. In contrast, the otariids swim with a high degree of agility and maneuverability using their fore flippers to generate thrust (e.g. refs. [27–33]) and they walk on land using all fours in quadrupedal gaits[30]. Therefore, the axial skeleton of otariids is much more flexible than that of phocids and, in general, the otariids present more developed hypaxial musculature than epaxial musculature, particularly at the posterior region of the column[31]. On the other hand, the walrus (*Odobenus rosmarus*), the only living odobenid, can perform these two types of aquatic locomotion but their land locomotion is like that of otariids[34].

Even though the vertebral column of aquatic taxa is more involved in locomotion and flexibility than in terrestrial taxa with fully-appendicular locomotion, it remains unknown whether and how the pattern of evolutionary integration and modularity exhibited by fissipeds[16] has changed during the evolution of aquatic carnivorans—i.e., pinnipeds—in response to the new locomotory demands. Deciphering whether a change in the pattern of integration and modularity happened in the evolutionary history of pinnipeds is capital to understand the evolution and differentiation of pinniped locomotor styles. Phenotypic integration refers to trait covariation within an organism due to genetic, developmental, and functional relationships and, therefore, the study of trait integration provides a deeper understanding of how selection acts on multiple traits simultaneously, leading to coordinated changes in the phenotype[35–37]. On the other hand, modularity refers to the presence of relatively independent modules within an organism and, hence, the study of modularity allows to identify the boundaries and interactions between different sets of functional traits, which can have important implications for evolvability and adaptive evolution[35–37].

In this study, we explore patterns of integration and modularity in the vertebral column of pinnipeds *sensu* the study of Martín-Serra et al.[16] for fissipeds. Our main goal is to investigate the impact of the land-to-water transition on patterns of axial integration and modularity in the vertebral column of pinnipeds related to their new locomotory demands. Specifically, we investigate: (1) changes in evolutionary integration and modularity patterns between the vertebral column of pinnipeds and the one previously obtained for fissipeds; (2) whether patterns of integration and modularity in the vertebral column of pinnipeds are associated to their new locomotory demands; and (3) differences between the patterns of integration in the vertebral column of phocids and otariids in relation to their different locomotor strategies on land and underwater. We use 3D geometric morphometrics to quantify shape covariation among presacral vertebrae in pinnipeds (Carnivora; Pinnipedia) and compare it with the results obtained for fissipeds by Martín-Serra et al.[16]. Moreover, to provide further evidence of the association between integration patterns and locomotory function, we CT-scanned a set of carnivoran species to assess for differences on the proportion of hypaxial and epaxial muscles across the presacral spine. Changes on these muscle bundles are thought to be closely related with differences in vertebral column mobility[31].

We hypothesize that the vertebral column of pinnipeds will be less integrated than that of fully-terrestrial taxa because they locomote less efficiently on land. We also predict that modularity patterns in the backbone of pinnipeds have changed in relation to the patterns exhibited by fissipeds because the vertebral column of aquatic taxa is more involved in locomotion and flexibility than in terrestrial taxa with fully-appendicular locomotion. Finally, we also hypothesize that phocids and otariids also present different patterns of integration and modularity because while phocids use

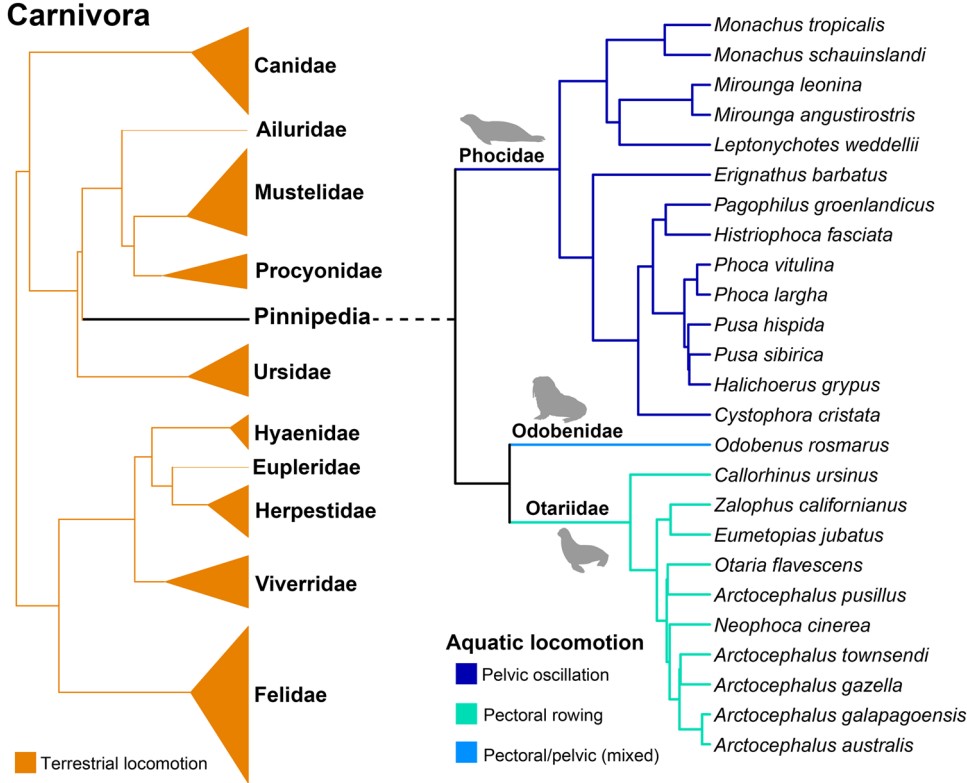

**Fig. 1 Phylogenetic tree topology used in comparative analyses for fissipeds and pinnipeds.** The tree topology and branch lengths are taken from ref. [48]. The swimming styles of crown pinnipeds are established following ref. [30]. All illustration are from PhyloPic (www.phylopic.org).

pelvic oscillation to generate thrust, otariids use their fore-flippers to generate underwater propulsion. Given that the axial musculoskeletal system is key for directing mobility and transmitting forces during locomotion[31], proportional differences of hypaxial and epaxial muscles between both groups of pinnipeds should also be noted.

## Results

**All pinnipeds**. The results obtained from the Procrustes ANOVAs performed for each vertebra, including all approaches of homology, indicate that, for most of them, the allometric effect was significant (Supplementary Data 2–5). Therefore, we used the residuals as size-free shape coordinates in subsequent analyses.

The results of the 2B-PLS indicated that the covariation between vertebrae was statistically significant across the whole vertebral column (with very few exceptions; see Supplementary Data 6–9) regardless of the count procedure. However, the strength of this integration (Z-scores) was not uniform across the vertebral column (Fig. 2). Even though there was no apparent pattern in the non-phylogenetic analyses, it arises with phylogenetic Z-scores for the four count procedures (Figs. 2 and S1). Thoracic vertebrae are integrated (TL01 to TL16 approximately), whereas the integration within cervical and lumbar vertebrae was comparatively weaker (Fig. 2a). Regarding the integration between different regions, cervical vertebrae were the weakest integrated with thoracic and lumbar ones (Fig. 2a). From the *thoracolumbar boundary count* and *selected vertebrae* procedures for all pinnipeds, it could be observed that the first lumbar is highly integrated with the thoracic vertebrae (Figs. S1c and 2a, respectively). The *diaphragmatic start count* procedure indicated a decrease of integration at PosD04-PosD05, which coincides with the thoracolumbar boundary (Fig. S1b). The *p* values of the differences between Z-scores from the 2B-PLS analysis indicated the absence of clear modules in the vertebral column of

pinnipeds (Tables 1–3)—although the strength of integration was remarkably high for specific sets of vertebrae (Fig. 2).

**Phocids**. The results from the Procrustes ANOVAs for each vertebra showed that allometric effect was significant for many vertebrae (Supplementary Data 10–13). Therefore, the residuals obtained were used as size-free shape coordinates for all vertebrae in the subsequent analyses.

The results of the phylogenetic and non-phylogenetic 2B-PLS and Z-scores indicated that the covariation between vertebrae was statistically significant for most of them (Supplementary Data 6–9) for all count procedures. In a similar way than for the previous analyses, the strength of integration was more evenly distributed across the vertebral column for non-phylogenetic analyses than for phylogenetic ones, in which a more integrated thoracic region could be observed for all count procedures (Fig. 2c). Again, the *diaphragmatic start count* procedure indicated a decrease of integration at PosD05, which corresponds with the thoracolumbar boundary (Fig. S1d). Although there were highly integrated sets of vertebrae across the column (Fig. 2), the *p* values of the differences between Z-scores from the 2B-PLS analysis were not significant (Tables 1–3).

**Otariids**. The results from the Procrustes ANOVAs for each vertebra indicate that, in otariids, the allometric effect were not significant for any vertebra (Supplementary Data 14–17). However, in order to ensure that both results were comparable between phocids and otariids (i.e., inflation of integration results), we used the residuals as size-free shape coordinates for all vertebrae of otariids, regardless of the non-significant association between allometry and vertebral shape in otariids.

The results of the phylogenetic and non-phylogenetic 2B-PLS and Z-scores indicated that the covariation between vertebrae was

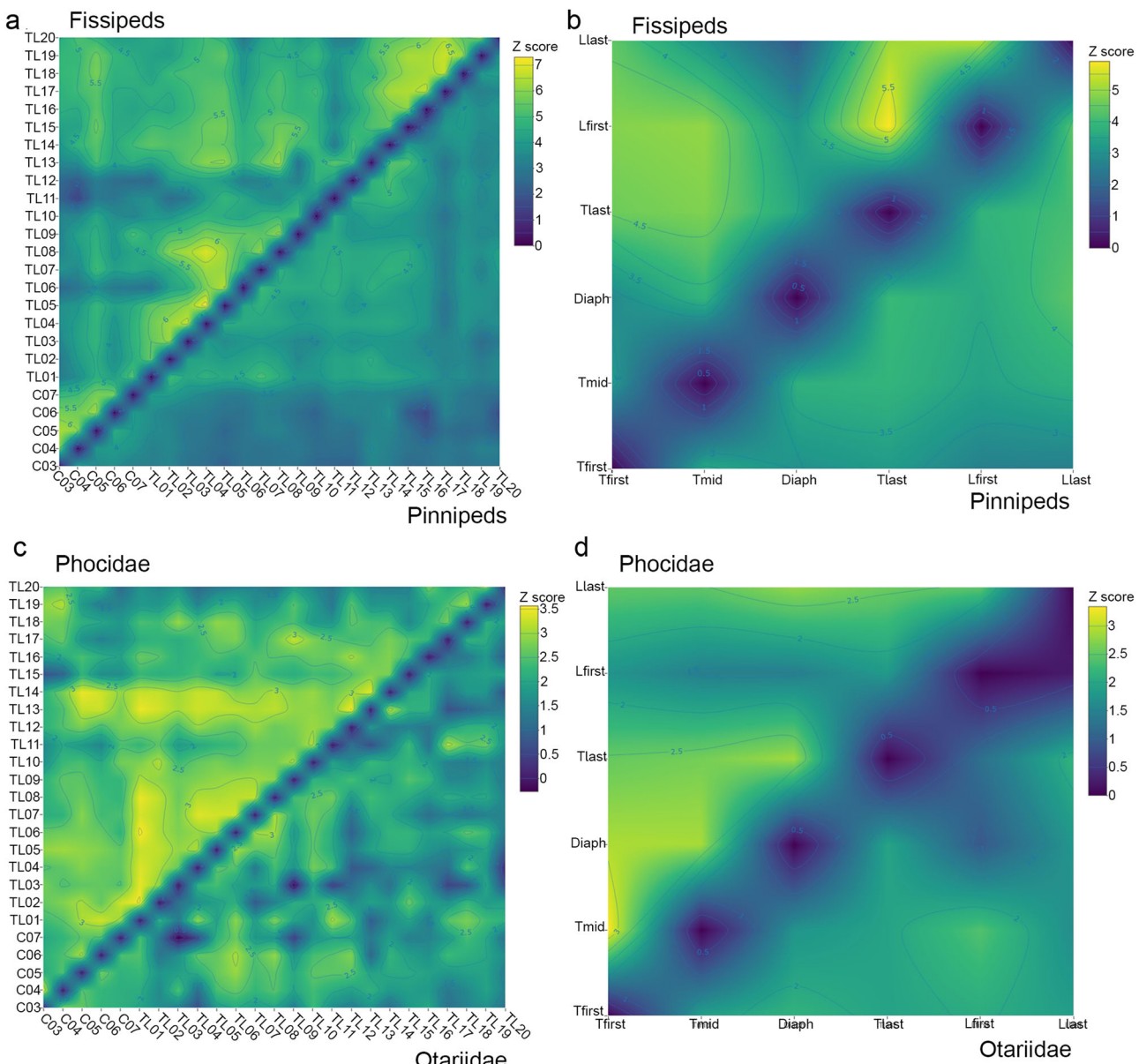

**Fig. 2 Heatmaps showing the strength of integration between each pair of presacral vertebrae. a** Strength of integration of fissipeds and crown pinnipeds using the *joined thoracolumbar count* procedure. The upper semimatrix corresponds to the *Z*-scores of fissipeds, all taken from Martín-Serra et al.[16] and the lower semimatrix to the *Z*-scores of pinnipeds obtained in this study. Following Martín-Serra et al.[16], the fissiped *Z*-scores were computed from size-residuals of vertebral shape. **b** strength of integration of fissipeds and pinnipeds using the *selected-vertebrae count* procedure. The upper semimatrix corresponds to the *Z*-scores of fissipeds taken from Martín-Serra et al.[16], and the lower semimatrix to the *Z*-scores of pinnipeds obtained in this study. **c** Strength of integration of phocids and otariids using the *joined thoracolumbar count* procedure. Upper and lower semi-matrixes correspond to *Z*-values of phocids and otariids, respectively. **d** Strength of integration of phocids and otariids using the *selected vertebrae* procedure. Upper and lower semi-matrixes correspond to *Z*-values of phocids and otariids, respectively. For the results obtained using other count procedures see Supplementary Data 3, 4 and Fig. S1.

statistically significant for many of them (Supplementary Data 6–9) in all count procedures. In general, the phylogenetic analyses of integration show an evenly integrated vertebral column with just subtle differences between regions (Figs. 2c, d and S1c, d). Likewise, it should be noted that the *p* values of the differences between *Z*-scores from the 2B-PLS analysis indicated the absence of modules across the column of otariids (Tables 1–3).

**Assessment of relative development and distribution of axial muscles**. Figure 3 shows the distribution of hypaxial and epaxial

muscles and their surfaces across the axial system of a fissiped (*P. lotor*) and two pinnipeds, one phocid (*P. vitulina*) and one otariid (*Z. californianus*). It is noteworthy the differences in distribution and arrangement of both hypaxial and epaxial muscles between regions. Comparative analysis of the epaxial and hypaxial muscle bundles along the vertebral column between fissipeds (Fig. 3a) and pinnipeds (Fig. 3a, c) showed that changes in the orientation and distribution of these muscles were less conspicuous in pinnipeds, which indicated that, in pinnipeds, the distribution was more uniform than in fissipeds between the two muscle blocks. The CT-scanned phocid (Fig. 3b and Supplementary Data 20–22) had a greater development of the epaxials in the lumbar region

**Table 1 Results of the modularity test using the *joined thoracolumbar count*.**

| | All pinnipeds | | | | Phocidae | | | | Otariidae | | | |
|---|---|---|---|---|---|---|---|---|---|---|---|---|
| | C07 TL05 | C07 TL15 | TL05 TL16 | TL15 TL16 | C07 TL05 | C07 TL14 | TL05 TL16 | TL14 TL16 | C07 TL06 | C07 TL14 | TL06 TL15 | TL14 TL16 |
| TL06 TL07 | - | - | - | - | - | - | - | - | 0.250 | - | 0.760 | - |
| TL05 TL07 | 0.483 | - | 0.990 | - | 0.330 | - | 0.570 | - | - | - | - | - |
| TL13 TL14 | - | 0.920 | - | - | - | 0.610 | - | 0.770 | - | 0.140 | - | 0.360 |
| TL13 TL15 | - | - | - | 0.380 | - | - | - | - | - | - | - | - |

*p* values of the comparisons between the Z-scores obtained from standard within-region comparisons (columns) and boundary comparisons (rows) for the cervicals and the thoracolumbars.

and an expansion of the hypaxial muscle in the thoracic region in comparison to otariids (Fig. 3c). The thoracic region of the otariid (Fig. 3c) was more similar to that of the fissiped. Moreover, a visual inspection of the relative distribution (i.e., the direction of bundles) of both hypaxial and epaxial muscles seem to be more uniform across spinal regions in otariids than in phocids.

## Discussion

Our results indicate that the presacral vertebrae of pinnipeds are integrated but the strength of integration is weaker than the integration exhibited by fissipeds (Fig. 2a) and quantified by Martín-Serra et al.[16]. Indeed, the Z-scores obtained from between-vertebrae comparisons range from 0.5 to 4.5 in pinnipeds, but from 1.0 to 7.0 in fissipeds (see Fig. 2a and Supplementary Data 2–5, 10–18). Based on this, we conclude that the axial skeleton of pinnipeds is less integrated when compared to their related terrestrial taxa. The strength of integration also changes between the two families of extant pinnipeds (i.e., phocids and otariids). Most of the between-vertebrae comparisons were significant in phocids, but few comparisons are significant in otariids (Fig. 2c and Supplementary Data 6–9). This indicates that the vertebral column of otariids is much less integrated than that of phocids.

Strikingly, modularity analyses also demonstrate that the three functional modules existing in the vertebral column of fissipeds are no longer present in pinnipeds, at least to a significant extent. This seems to be supported by our assessment of the relative development of hypaxial and epaxial muscles and the orientation of their bundles, because, in fissipeds, changes in these muscles are larger and more abrupt across axial regions than in pinnipeds (Fig. 3). However, it is worth mentioning that we have compared these parameters to only one fissiped, which could represent a specific pattern of this species, instead of being the general condition for all fissipeds. In any case, future studies based on CT-data will confirm or refute our generalizations.

However, it is worth mentioning that the lower variation in the Z-score obtained from between-vertebrae comparisons of pinnipeds relative to fissipeds may bias the significance of the modularity tests—i.e., pairwise comparisons between the Z-scores of vertebral pairs. Indeed, using the *joined thoracolumbar count* procedure, our results demonstrate that pinnipeds possess a set of thoracic vertebrae (~TL01-TL16) that are integrated compared to other vertebrae of the presacral spine (Fig. 2c). Moreover, this relatively integrated thoracic segment is accompanied by two weakly-integrated sets of cervical and lumbar vertebrae. In addition, the diaphragmatic vertebra does not appear to be as crucial in pinnipeds as it is in fissipeds since it is integrated within the rest of the vertebrae of the thoracic segment. Therefore, the separation of both dorsal modules (i.e., anterodorsal vs. posterodorsal) by the de-integrated diaphragmatic vertebra that characterize the presacral column of fissipeds is, to some extent, blurred in pinnipeds. We hypothesize that these changes may be due to the functional demands of the new physical environment they inhabit. It has been suggested that the presence of anterodorsal and posterodorsal modules serve to avoid compromising trunk ventilation and exceeding the extension of the posterior back while locomoting on land[16]. Thus, the diaphragmatic vertebra in fissipeds may act as a hinge between the motion-restricted thoracic vertebrae with vertebrosternal ribs and the more mobile thoracic vertebrae with floating ribs plus the lumbar series[16,20]. We hypothesize that, in pinnipeds, the release (in part) of the trade-off between respiration vs. locomotion may also imply that the diaphragmatic vertebra is no longer required to act as a hinge between the anterodorsal and posterodorsal module and, therefore, is relatively integrated into the "thoracic" segment

**Table 2 Results of the modularity test using the *thoracolumbar boundary count*.**

|  | All pinnipeds | | Phocidae | | Otariidae | |
|---|---|---|---|---|---|---|
|  | T12 L01 | T05 L01 | T03 L01 | T02 L01 | T07 L01 | T04 L01 |
| T03 T02 | – | – | 0.752 | 0.662 | – | – |
| T07 T04 | – | – | – | – | 0.082 | 0.110 |
| T07 T05 | – | 0.863 | – | – | – | – |
| T12 T02 | 0.828 | – | – | – | – | – |

*p* values of the comparisons between the *Z*-scores obtained from standard within-region comparisons (columns) and boundary comparisons (rows) for the cervicals and the thoracolumbars.

**Table 3 Results of the modularity test using the *diaphragmatic start count*.**

|  | All pinnipeds | | Phocidae | | Otariidae | |
|---|---|---|---|---|---|---|
|  | PreD02 PostD04 | PostD04 PostD05 | PreD01 PostD04 | Diaph PostD04 | PreD02 PostD05 | PostD03 PostD04 |
| PreD01 Diaph | – | – | 0.592 | 0.915 | – | – |
| PreD02 PostD02 | – | – | – | – | 0.904 | 0.356 |
| PreD02 PostD03 | 0.551 | 0.250 | – | – | – | – |

*p* values of the comparisons between the *Z*-scores obtained from standard within-region comparisons (columns) and boundary comparisons (rows) for the cervicals and the thoracolumbars.

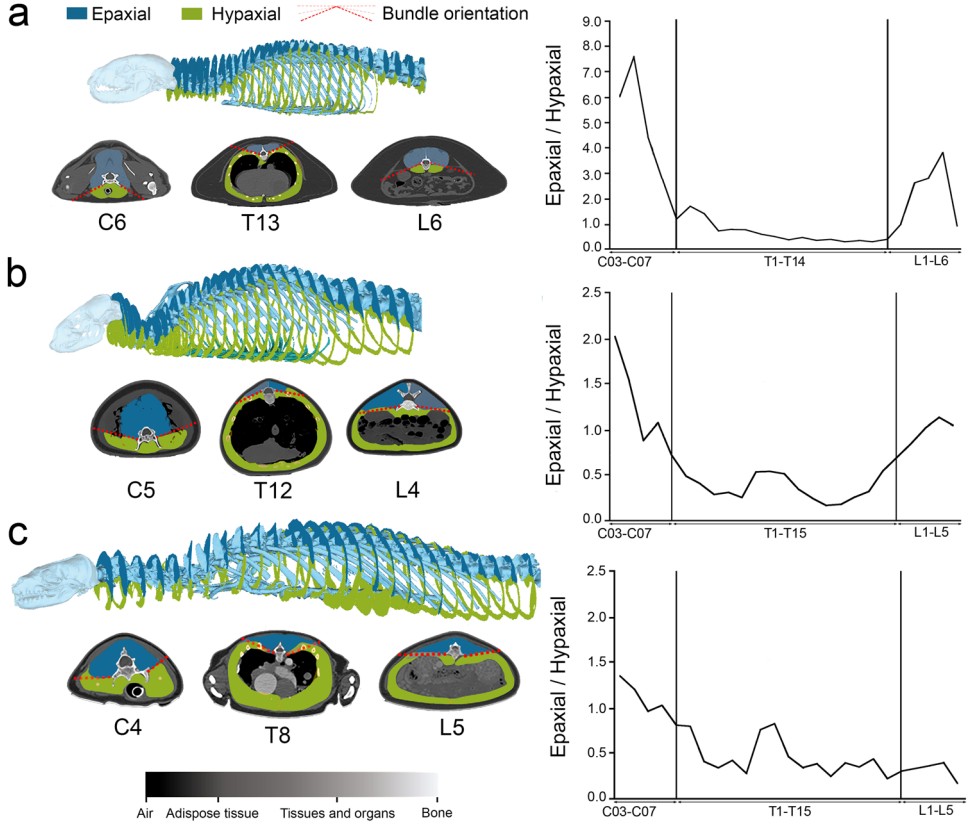

**Fig. 3 Muscle anatomy of fissipeds and pinnipeds. a** Sagittal slices of a specimen of *Procyon lotor*. **b** Sagittal slices of a specimen of *Phoca vitulina*. **c** Sagittal slice of a specimens of *Zalophus californianus*. Dashed red lines represent the boundary between epaxial and hypaxial muscles beginning on the horizontal septum and indicate the relative orientation of bundles. Bivariate graphs represent the ratio between the surfaces of epaxial muscles relative to hypaxial muscles (both in mm$^2$) against the position of each vertebra.

that characterizes the integration pattern of the presacral vertebrae. One of the reasons are that pinnipeds combine short breathing periods (eupnea) with longer-duration of breath hold periods (apnea)[38]. For example, *Neophoca cinerea* and *Phocarctos hookeri* typically breathe 3–5 times per minute[39]. Some elephant seals (i.e., *Mirounga*) exhibit a hyperventilation behavior of breathing during surfacing periods between dives at sea, breathing at a rate of ~15 breaths/min during 2–3 min[40]. Therefore, pinnipeds neither breathe at the same rate than fissipeds nor breathe while locomoting on land or in water. This could relate to

the cost of transport (i.e., the power required to move a given body mass at some velocity) of pinnipeds in water, which is lower than that of semi-aquatic fissipeds such as the North American mink (*Neovison vison*)[41]. Indeed, swimming is the least costly mode of locomotion among the general modes of transport in animals[42]. This is related to the fact that swimmers do not need to support their body weight against the constant pull of gravity[43]. However, it is striking that pinnipeds appear to have similar breathing patterns on land as they do at sea[44], because they combine short breathing periods with longer-duration of breath hold periods[38]. Moreover, the integration of the diaphragmatic vertebrae into the "thoracic" segment of pinnipeds could also be related to the fact that the mammalian asymmetrical gaits typical of sagittal locomotion are not as important in pinnipeds as in fissipeds. Indeed, the postdiaphragmatic vertebrae are key for sagittal bending during asymmetrical gaits in terrestrial mammals[14].

Our analyses of integration and modularity performed separately for phocids and otariids indicate that phocids possess a pattern of integration like those obtained from the complete sample: a relatively integrated "thoracic" segment (including the diaphragmatic vertebrae) but with de-integrated cervical and lumbar regions (Fig. 2b). In contrast, otariids lack the relatively integrated "thoracic" segment exhibited by phocids (Fig. 2c). This seems to be supported by CT-scan data, because the proportion of hypaxial and epaxial bundles are more homogeneous across spinal regions in otariids than in phocids (Fig. 3b, c and Supplementary Data 20–22), which may indicate that the presacral spine of otariids is a functional unit. However, our limited CT-dataset could represent interspecific differences instead of differences between phocids and otariids.

The slightly integrated "thoracic" segment with de-integrated lumbar and cervical regions that characterizes the pattern exhibited by phocids may be related to their locomotor style. Phocids are hindlimb-dominated swimmers that use pelvic oscillation to generate forward thrust, a behavior in which the anterior body is held rigid and the lumbar region is subject to lateral undulations coupled with lateral sweeps of the hind flippers[25–28]. Furthermore, their thoracic region is rigid, while their lumbar area is very flexible with large epaxial muscles providing the necessary movement to the lumbar region[31] with a massive and large sacrum[22]. Indeed, our CT-scan data support this, as the direction and orientation of the epaxial and hypaxial bundles along the lumbar series are similar in the three specimens (*P. lotor*, *Z. californianus* and *P. vitulina*) analyzed in Fig. 3, but with great differences in the relative development of these muscles. Phocids have much more developed epaxial muscles (hindlimb-dominated swimmers) than the otariid and the fissiped (Fig. 3). Therefore, it seems that the relatively-integrated, and possibly motion-restricted, thoracic segment of phocids is associated with a de-integrated and highly-mobile lumbar series. Phocids undulate their half to posterior portion of their bodies to generate thrust and keeping the anterior portion steady but having maneuverability[30].

Similarly, the lack in otariids of the relatively integrated "thoracic" segment of phocids (Fig. 2) could be related to the high flexibility that characterize their spines. This also applies to the lack of cervical and lumbar modules. Indeed, otariids are considered forelimb-dominated swimmers because they use their fore flippers to generate thrust and, although their hindlimbs and the vertebral column play no apparent role in generating propulsion (e.g. refs. [27–30]), their spines are characterized by having very flexible intervertebral joints, suggesting a link between the axial skeleton and the improved maneuverability and turning[31]. Indeed, CT-scan data indicates that the orientation of hypaxial and epaxial bundles are almost similar across the three regions in

the otariid and both blocks of muscles are also individualized along the spine, which could relate to their improved maneuverability and turning.

Our morphometric data suggest a link between axial flexibility and the different integration patterns found between fissipeds vs. pinnipeds and between otariids vs. phocids. More integrated axial segments may be related to regions of restricted motion and segments with a reduced (or even absent) integration seem to be highly-mobile regions. Moreover, our CT-scan data seem to confirm this relationship because the proportion of epaxial and hypaxial muscles across the spine relate to different locomotor strategies. Future comparative studies of intervertebral joint mobility among these taxa could further clarify if there is a direct link between the strength of integration and both spinal flexibility and maneuverability in pinnipeds.

In any case, here, we document a significant impact of the land-to-water evolutionary transition on the evolution of the mammalian backbone, reducing the integration typical of terrestrial taxa and lacking the underlying functional modules to vertebral regionalization of this multi-element and serially homologous structure that characterize the vertebrate body plan.

## Methods

We scanned 1075 presacral vertebrae (from C03 to the last lumbar) using a surface scanner EinScan Pro 2X belonging to 44 specimens of 28 species (Table S1) with a range of 1–4 specimens per species, depending upon availability in museum collections. Therefore, an average of 25 vertebrae per specimen was digitized. However, one of the sampled specimens belonging to *Hydrurga leptonyx* had a missing vertebra, which forced us to remove this species from the analyses. All specimens were adults following fully-closed basilar synchondrosis and complete fusion of epiphyses to diaphysis. We also made efforts to sample equal numbers of males and females per species when this information was available in museum collections (see Table S1). A series of 40 homologous 3D landmarks were digitized in all these vertebrae using the software *Stratovan Checkpoint*[45] to capture their main morphological features (Fig. 4 and Table S2).

We categorized all vertebrae based on their respective position. Nevertheless, since the number of thoracolumbar vertebrae varies across species, we used the procedures described by Martín-Serra et al.[16] to conform with different hypotheses of homology: (1) *the joined thoracolumbar count* in which we joined thoracic and lumbar vertebrae into a single region; (2) *the thoracolumbar boundary count*, beginning to count vertebrae at the thoracolumbar boundary and tallied lumbar vertebrae in a caudal direction and thoracic vertebrae in a cranial direction; (3), *the diaphragmatic start count*, which uses the diaphragmatic vertebrae as the starting point for counting in both caudal and cranial directions. Finally, we also applied the *selected vertebrae* procedure, analyzing the first, middle, and last thoracic vertebrae, the diaphragmatic vertebra, and the first and last lumbar vertebrae.

The *x*, *y*, *z* landmark coordinates (Supplementary Data 1) were uploaded to R environment using the *geomorph* (v. 4.0.5) package[46]. Subsequently, a phylogeny of the species included in this sample (Fig. 1) was assembled using *ape* package[47]. This phylogeny was built following the one published by Nyakatura and Bininda-Edmons[48].

We performed a Procrustes superimposition[49] for each vertebral position accounting for bilateral symmetry using the *geomorph* (v. 4.0.5) package[46]. We averaged the resulting Procrustes coordinates of each vertebra for those species represented by more than one individual. Afterwards, we performed a phylogenetic Procrustes ANOVA (PGLS[50]) with log-transformed

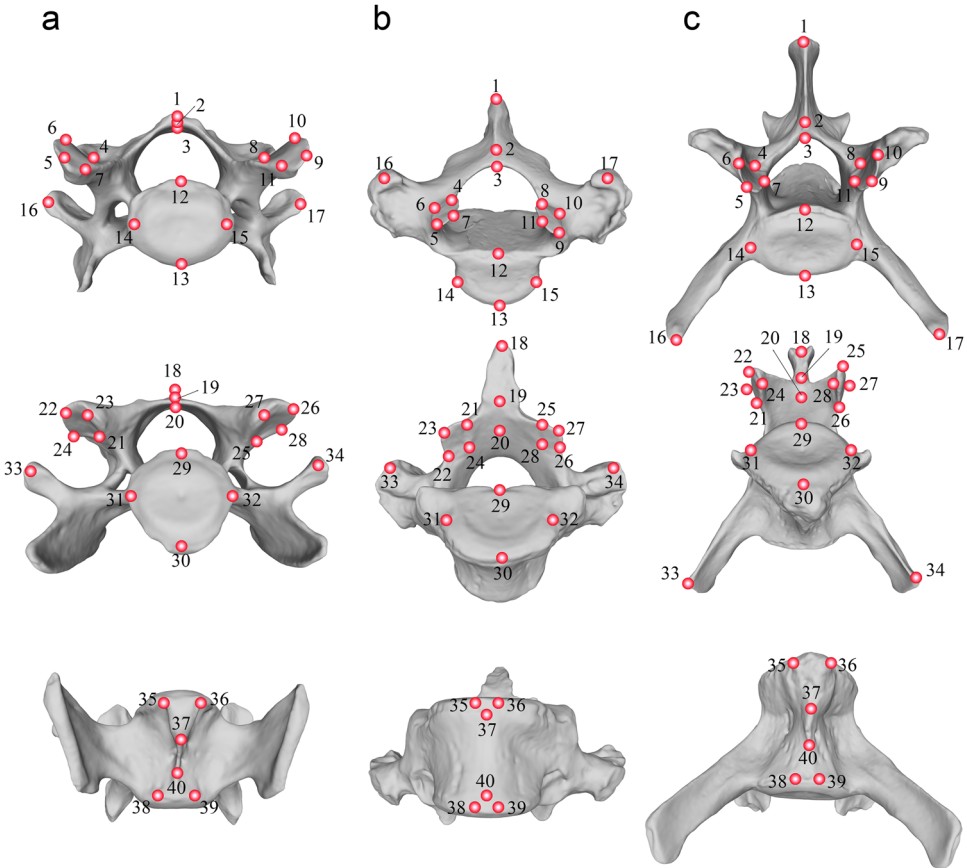

**Fig. 4 Three-dimensional landmarks digitized in the presacral vertebrae of pinnipeds.** See also Table S2 for anatomical criteria. **a** Cervical vertebrae. **b** Thoracic vertebrae. **c** Lumbar vertebrae. Only a vertebra for each regional type is shown for clarity. The vertebrae belong to the species *P. vitulina*.

centroid size as independent variable to test for allometric shape variation using the *geomorph* (v. 4.0.5) package[46]. If allometry was significant, we used the residuals as size-free shape coordinates for all subsequent analyses. All these analyses were repeated for the four procedures described above.

We carried out the analyses of integration for each count. The strength of integration between each pair of vertebrae was estimated using the shape covariation computed by phylogenetic and non-phylogenetic two-block Partial Least Squares (2B-PLS) and the *Z*-score[51] using *geomorph* (v. 4.0.5)[46]. The *Z*-score was used as a measure for the extent of integration between pairs of vertebrae[51]. To compare the integration strength (*Z*-score) obtained for pinnipeds in this study with that obtained by Martín-Serra et al.[16] for fissipeds, we repeated the analyses of integration after removing the landmarks digitized on the transverse processes of lumbar vertebrae (L16, L17, L33, L34) and on the ventral part of the centrums (L35–L40). Moreover, we used the Benjamini-Hochberg method[52] to avoid false rejections of null hypothesis for each set of 2B-PLS. We used the *compare.pls* function to make a statistical comparison between the effect sizes, *Z*-score, of multiple PLS analyses[46]. Its main application was to evaluate the levels of integration between different modules in various samples. The statistical significance of the difference between effect sizes is evaluated using the pooled standard error from the sampling distributions. We took the highest value of *Z*-score inside a given module (among all comparisons, i.e., between adjacent and non-adjacent vertebrae) obtained from 2B-PLS analyses and we compared it with the vertebrae located at the hypothetical inter-module boundaries (i.e., breaks in the levels of integration). Changes in the *p* value obtained from such comparisons, transitioning from non-significant to significant,

confirms the boundary location. Note that, using this procedure, there is not a single threshold in the differences values that marks the level of significance, as it can be different among comparisons[16]. All statistical analyses were performed for three different grouping of the data: all the specimens, only phocids, and only otariids. We were not able to perform the same analysis for odobenids because there is only one living species.

To explore whether the patterns of axial integration and modularity relate to the distribution and development of hypaxial and epaxial muscles in pinnipeds, we used CT scans (see Supplementary Data 19 for parameters of data acquisition) of captive animals from the Oceanographic Park (Valencia, Spain) and from the Zooparc de Beauval in France (ZPB). Specifically, we used a specimen of *Phoca vitulina* (OV PV 001) and a specimen of *Zalophus carlifornianus* (ZPB ZC 004). Moreover, we also used a CT-scan of *Procyon lotor* (ZPB_PL_003) as an example of a fissiped. All animals were anaesthetized and scanned for routine healthcare tests and not specifically for this investigation. Our purpose was to investigate if there are differences in muscle arrangements at key regions across the spine by computing the ratio between the surface area of epaxial and hypaxial muscles (see Supplementary Methods for more details).

**Reporting summary**. Further information on research design is available in the Nature Portfolio Reporting Summary linked to this article.

## Data availability
The landmarks coordinates are available as Supplementary Data 1.

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

## Acknowledgements

We are grateful to Sara Ketelsen from the American Museum of Natural History, Daniel Klingberg from the Zoological Museum of the University of Copenhagen (Denmark), Cécile Blondel from the Muséum National d'Histoire Naturelle of Paris, Jorge Vélez-Juarbe from the Natural History Museum of Los Angeles County, and Carlos Rojo from the Oceanogràfic of Valencia (Spain) for kindly providing access to the specimens under their care through project number OCE-06-21. This study has been funded by the Spanish Ministry of Economy and Competitiveness (MINECO) (grant numbers CGL2015-68300P and PID2019-111185GB-I00) and Junta de Andalucía (P18-FR-3193) to B.F.

## Author contributions

J.M.E. collected, analyzed, interpreted data and wrote the paper; A.M.S. analyzed data, contributed in data collection, interpreted data, and wrote methods and results; A.P.R. analyzed the CT-scans, contributed in data collection, interpreted the results of the myology section, and wrote this part of manuscript; B.M. contributed with CT-scans; K.J. contributed in results interpretation and discussion, and assisted with writing; B.F. conceived and coordinated the study, contributed in data collection, wrote the paper and get funds.

## Competing interests

B.F. is an Editorial Board Member for *Communications Biology*, but was not involved in the editorial review of, nor the decision to publish this article. The other authors declare that there are no competing interests.
