## [Peer Review File · Communications Biology]

Reviewers' comments:

Reviewer #1 (Remarks to the Author):

The authors employed 3D Geometric morphometrics to investigate the integration of vertebral shape along the column of Pinnipeds, and compare it to a previous work done on Fissipeds.

The work is original and relevant to the field, since there are not many works dealing with 3D morphology of the post cranium of marine mammals. I congratulate the authors for their efforts. In an attempt to aid the authors, I have made over forty comments for minor changes along both the manuscript and the Supplementary methods file. There are also some suggestions with minor edits of the language.

The employed methods are solid and support the discussion to a great extent although some clarifications are required to improve the discussion. I consider that sample size is large enough, although there are no clarifications about the ontogenetical stage of the specimens or their sex (see comment on file); two factors that could influence the results due to sexual segregation in some species. My only concern with the data set is the presence of *H. leptonyx* and how its locomotor style could be influencing results on phocids (See comment in the discussion section).

In addition to this, I feel that the M&M section could be greatly improved by providing references on the digitized landmark (adding number to the figure, see comment on Figure 1 and Table S2). Even though the authors have done a good job in writing the manuscript, I feel that the style and language could be greatly improved and I have made my best to suggest possible grammar and style improvements. Both the Introduction and Discussion need some work regarding coherence and cohesion, especially the former.

Particularly, regarding language, some sentences need re-writing for the sake of clarity. On what style regards, there are multiple minor consistency issues (that I have pointed out throughout the text and the Supplementary Materials) such as: indentation, terminology, style of statistical reports (especially p- values and z-scores).

What has surprised me is that I haven't been able to find Figure S1, which was numerously referred to in the text (maybe I am missing something).

Overall, the manuscript is a robust work, contributing enormously to its field. Despite this, I feel that it would benefit from thorough revision. I look forward to reading its final version.

Reviewer #2 (Remarks to the Author):

The manuscript describes the impact of secondary aquatic adaptations and swimming modes on the integration along the vertebral column of pinnipeds. The authors employed geometric morphometric analysis on an impressive dataset (1075 vertebrae analyzed), and also quantified the muscle arrangement along the axial skeleton in three species (one fissiped, one otariid and one phocid). The study reveals that the axial skeleton of pinnipeds exhibits lower integration compared to fissipeds. Furthermore, they observe that more integrated vertebral segments correspond to regions with limited motion, a finding also endorsed by the musculature data.

Although the manuscript is promising to elucidate many interesting questions, there are several aspects that require considerable attention and restructuring. Particularly, the manuscript needs to be revised for i) clarifying the objectives and ii) ensuring the reliability of the results. I found it difficult to follow the objectives, as some points of the introduction fail to communicate with the proposed methods. Lack of clarity is also present in the methods, results and in certain figures (see specific comments below). Additionally, some of the results need further confirmation, as certain key findings are presented with non-comparable data.

i) Clarification of objectives

1) Introduction

The introduction doesn't fully capture the innovation of the work, and fails to clearly state all the objectives pursued by the authors. The manuscript's goals are diluted along the text, and these are first presented as a continuation to a previous work (L 76). In this current format, the importance of the ecological aspects that make pinnipeds the ideal system for such investigation is underemphasized. Specifically, the theoretical background presented from the third paragraph onward needs to be better connected to enhance the text's overall comprehensibility. Additionally, an explanation of the importance of modularity and integration in phenotypic evolution is missing, which would assist readers in understanding the authors' motivations. These aspects are somehow already written in the introduction, but need to be restructured for to better present the objectives. It is also crucial to include a final paragraph clearly stating all the objectives and hypotheses, which is inexistent in this current form.

2) Methods

In their methods, the authors introduce an unexpected objective that deviates from what was described in the introduction. They argue that they used CT scans to compute the ratio area of epaxial and hypaxial muscles, and relate this information to the integration of the axial skeleton. However, the introduction does not explain the significance of such muscles in vertebral function or why they should be considered in investigating axial skeletal integration. Additionally, there is no mention of how many species or individuals were analyzed, or how many vertebrae per individual. These elements need to be presented.

ii) Reliability of results

3) The authors identified differences in the z-scores between phocids and otariids, with phocids showing higher integration than the latter. However, the type of data used to calculate such scores is not the same for both groups: in phocids, they used size residual scores, and in otariids, they used the superimposed coordinates. It is unclear whether the fissipeds used size residuals or superimposed coordinates since it is not mentioned. This is an important issue because using size residuals in one group but not in the other, regardless of the non-significant association between allometry and vertebral shape in otariids, may have inflated the integration results. The authors must ensure that both variables are comparable between groups and standardize the data to confirm the reliability of their findings.

Specific comments:

4) L 48-50. Include a citation for this sentence. Wellik & Capecchi (2003) are cited later at the end of the phrase, but their work did not provide a characterization of backbone regionalization in "land-going vertebrates".

5) L 61: correct "has" to "have".

6) L 61. "That" needs a complement here.

7) L 63-70. Simplify and split the sentence for clarity. Example: "The last two modules likely relate to locomotor performance, as they appear to be related to motion capability of the presacral spine 16. While the vertebrae of the anterodorsal module (...) from the 'Diaphragmatic joint complex' 17-20."

8) L70-75. These sentence confusing, please clarify it.

9) L 80. "Effects of gravity" are not absent underwater. Instead, animals moving in this medium endure gravity and buoyancy. Rephrase for precision.

10) L 81-83. Clearly state that such a situation only applies while underwater since aquatic tetrapods can likewise spend time on land and support their weight/ move by generating friction with the substrate.

- 11) L 83-85. Improve the sentence so to better linked it to the previous information. Example: In the axial skeleton of some marine mammals, such as cetaceans, secondary aquatic adaptations involved the reduction of regionalization including (...).
- 12) L 76-79. This sentence here weakens the novelty of your work by stating that is a continuation of previous studies but on a different clade. I believe that it should be presented only after providing the state-of-the-art on the integration and modularity of the vertebral column in pinnipeds. By doing so, the introduction now highlights the innovation of addressing these questions in aquatic carnivores, making your biological question much more interesting. I recommend moving this sentence to a new last paragraph of your introduction, where your goals are clearly stated.
- 13) L 90-93. The use of "by integrating" here is odd. Please rephrase the sentence. One possible alternative: "demonstrate a reduction in the number of modules in which lumbar, sacral, and anterior caudal are integrated into a single (...)"
- 14) L 95-97: Repetition of lines 86-89, exclude or restructure paragraphs.
- 15) L 98: fix misspelling in 'less'
- 16) L 97-100. This should also be better placed in a last paragraph of objectives and hypothesis.
- 17) L101-107. This paragraph is lost here. It's better suited after L 89.
- 18) L 109: This count represents, in average, how many vertebrae per species? Also, precise how many species you used and the range of individuals/ sp.
- 19) L 152: Were all fissipeds extracted from Martin-Serra et al 2021? Precise.
- 20) L 175-176: This premise came as a surprise, nowhere in the introduction you mention hypaxial and epaxial muscles. Make sure you present such assumptions in the introduction.
- 21) L 201-202: Where fissiped data comes from should be stated in the methods as well.
- 22) L 280 Figure 4. Remove "of".
- 23) L 298-300 This affirmation needs to be tempered. Comparing these parameters to only one fissiped species is not enough for such generalization.
- 24) L 304-305. Yes, but your parameters are not quite comparable. Se general comment 3.
- 25) L 305-307. Again, this affirmation needs to be tempered.
- 26) L 340: Correct to "swimmers do not need to support their (...)"
- 27) In the results: integration results of fissipeds are not described (just presented in Figure 3). Did you use size residuals or Procrustes coordinates? Precise.
- 28) Figure 1: What is A B and C? Precise which species, axial region, and anatomical orientation.
- 29) Figure 3. Increase axis text, and describe what the abbreviations represent. B) correct "phocids" to phocids. C. Isn't it "fissipeds" and "pinnipeds" instead of "phocids" and "otariids"? Use bars alongside the axes to indicate the region to which the vertebrae belong to.

Reviewer #3 (Remarks to the Author):

Summary:

This is an elegant morphological data set comparing vertebral column integration and surrounding musculature in seals and otters, and within the seals comparing the true seals and sea lions / fur seals. This paper uses specimens from collections to quantify morphometrics along the vertebral column and ct scans of living specimens to quantify epaxial and hypaxial musculature among groups. The authors found that while seals are integrated, they are less integrated than otters. Within the seals, the true seals are more integrated than the sealions/ fur seals. The authors discuss the functional modules or lack of functional modules in the different groups and eventually bring these ideas back to address locomotion and evolutionary transitions.

Major comments:

The morphology in this paper is really lovely, but this paper in its current form is very technically related to pinnipeds and fissipeds. I think the paper, the intro specifically, might need to be reframed to be more obviously addressing land to water evolutionary transitions and locomotion differences in these groups. I am not sure this current version would appeal to the broad readership of the selected journal. I know space is limited, but I think this paper would benefit from a schematic including a phylogeny of these groups showing how they are related and how their locomotion might differ. This would be much more meaningful to me than Tables 2 and 3, which show Z scores. I think these tables should be supplemental. Readers can glean these from the results section and figure 3.

Minor comments:

1; Line 29: define fissipeds

2; Line 63-70: This sentence is very long and difficult to follow. It might be best to break these ideas up for easier consumption by the reader.

3; Line 70 and 74: It would be useful to give an definition of the Diaphragmatic joint complex.

4; Line 81-83: For clarity, I suggest saying fully aquatic mammals (living entirely in water), which can contrast to partially aquatic mammals, or something similar, to refer to other species. As it is written, many of the animals talked about in the intro generate friction with the substrate when they are moving on land.

5; Line 86: maybe revise to say 'pinnipeds, partially aquatic mammals,...' These changes make evolutionary and function groupings clear to the readers of a large broad journal.

6; Line 98: 'less'

7; Line 101: should this list include 'and'?

8; Line 151: replace 'ref' with author's names

Results: It is confusing that Fig 3A and 3C are first compared. Consider re-lettering the figure so that you can present 3A and 3B first in the results section and then 3C and 3D.

9; Figure 4: This is really interesting and well presented.

10; Line 329: The hinge is a really cool idea! The following paragraphs are really interesting. This paper would benefit from moving some of these ideas to the intro as a functional hypothesis to be tested throughout this study.

REVIEWER #1 (comments in text highlighted in yellow)

The authors employed 3D Geometric morphometrics to investigate the integration of vertebral shape along the column of Pinnipeds, and compare it to a previous work done on Fissipeds. The work is original and relevant to the field, since there are not many works dealing with 3D morphology of the post cranium of marine mammals. I congratulate the authors for their efforts.

Thank you very much for your comments to our paper. We really appreciate them.

In an attempt to aid the authors, I have made over forty comments for minor changes along both the manuscript and the Supplementary methods file. There are also some suggestions with minor edits of the language.

Thank you very much for all corrections made and for such a constructive review.

The employed methods are solid and support the discussion to a great extent although some clarifications are required to improve the discussion. I consider that sample size is large enough, although there are no clarifications about the ontogenetical stage of the specimens or their sex (see comment on file); two factors that could influence the results due to sexual segregation in some species.

We have clarified this in Table S1 and in the method section (see lines 161-164 and new Table S1).

My only concern with the data set is the presence of *H. leptonyx* and how its locomotor style could be influencing results on phocids (See comment in the discussion section).

In the past version of the paper, we actually removed *H. leptonyx* from our database because this specimen lacks one vertebra and it is not possible to perform integration analyses with incomplete columns. Therefore, it was our mistake to do not indicate that despite *H. leptonyx* was sampled in our study, it was not possible to analyze this species. However, in the new draft of the paper, we have specified this. Lines 159-161 and see also the caption of Table S1.

In addition to this, I feel that the M&M section could be greatly improved by providing references on the digitized landmark (adding number to the figure, see comment on Figure 1 and Table S2).

We have included landmark numbers to Figure 1 (now Figure 2) and we have specified the morphological criteria used. See new Figure 2 and new Table S2.

Even though the authors have done a good job in writing the manuscript, I feel that the style and language could be greatly improved and I have made my best to suggest possible grammar and style improvements. Both the Introduction and Discussion need some work regarding coherence and cohesion, especially the former.

We have worked for improving coherence and cohesion, particularly across the introduction. See new introduction and discussion.

Particularly, regarding language, some sentences need re-writing for the sake of clarity. On what style regards, there are multiple minor consistency issues (that I have pointed out throughout the text and the Supplementary Materials) such as: indentation, terminology, style of statistical reports (especially p- values and z-scores).

Thank you. We have incorporated all your suggestions and we have read carefully the MS to look for additional inconsistencies. All your comments/suggestions/changes in the PDF have been incorporated into the main text highlighted in yellow.

What has surprised me is that I haven't been able to find Figure S1, which was numerously referred to in the text (maybe I am missing something).

We really apologize for the omission. We do not know how this has happened but now we have incorporated Figure S1 in the new supplementary material.

Overall, the manuscript is a robust work, contributing enormously to its field. Despite this, I feel that it would benefit from thorough revision. I look forward to reading its final version.

Once again, thank you very much for your constructive review.

REVIEWER #2 (comments in text highlighted in green)

The manuscript describes the impact of secondary aquatic adaptations and swimming modes on the integration along the vertebral column of pinnipeds. The authors employed geometric morphometric analysis on an impressive dataset (1075 vertebrae analyzed), and also quantified the muscle arrangement along the axial skeleton in three species (one fissiped, one otariid and one phocid). The study reveals that the axial skeleton of pinnipeds exhibits lower integration compared to fissipeds. Furthermore, they observe that more integrated vertebral segments correspond to regions with limited motion, a finding also endorsed by the musculature data. Although the manuscript is promising to elucidate many interesting questions, there are several aspects that require considerable attention and restructuration. Particularly, the manuscript needs to be revised for i) clarifying the objectives and ii) ensuring the reliability of the results. I found it difficult to follow the objectives, as some points of the introduction fail to communicate with the proposed methods. Lack of clarity is also present in the methods, results and in certain figures (see specific comments below). Additionally, some of the results need further confirmation, as certain key findings are presented with non-comparable data.

Thanks for the comments regarding our MS and for finding the time to perform such a constructive review. Below, we answer all reviewer concerns point-by-point.

1) Introduction. The introduction doesn't fully capture the innovation of the work, and fails to clearly state all the objectives pursued by the authors.

In the new version of the MS we have specified the goals of our paper (Lines 129-134).

The manuscript's goals are diluted along the text, and these are first presented as a continuation to a previous work (L 76).

Now, we have reformulated this sentence, and we are no longer saying that “we extend the paper of Martín-Serra et al. to pinnipeds”. Instead, we say that we analyze the evolutionary integration and modularity of the vertebral column of pinnipeds sensu Martín-Serra et al. (Lines 115-116).

In this current format, the importance of the ecological aspects that make pinnipeds the ideal system for such investigation is underemphasized.

We have highlighted the importance of the ecological aspects that make pinnipeds and ideal system for our investigation (Lines 89-109).

Specifically, the theoretical background presented from the third paragraph onward needs to be better connected to enhance the text's overall comprehensibility.

Accordingly, we have connected better the theoretical background of our study from the third paragraph with the onward text (see Lines 89-154). We hope to have improved the text's overall comprehensibility.

Additionally, an explanation of the importance of modularity and integration in phenotypic evolution is missing, which would assist readers in understanding the authors' motivations. These aspects are somehow already written in the introduction, but need to be restructured for to better present the objectives.

We have incorporated a paragraph across the introduction explaining the importance of modularity and integration in phenotypic evolution (Lines 115-128). We hope that this new paragraph will assist readers in understanding author's motivation.

It is also crucial to include a final paragraph clearly stating all the objectives and hypotheses, which is inexistent in this current form.

We have included both, the objectives (Lines 129-140) and hypotheses (Lines 141-154).

2) In their methods, authors introduce an unexpected objective that deviates from what was described in the introduction. They argue that they used CT scans to compute the ratio area of epaxial and hypaxial muscles, and relate this information to the integration of the axial skeleton. However, the introduction does not explain the significance of such muscles in vertebral function or why they should be considered in investigating axial skeletal integration.

We have included across the introduction a paragraph explaining why the epaxial and hypaxial muscles are important to understand locomotor performance in pinnipeds. See Lines 100-102; Lines 104-107; Lines 137-140; Lines 150-152.

Additionally, there is no mention of how many species or individuals were analyzed, or how many vertebrae per individual. These elements need to be presented.

This information was available in Table S1 of the original manuscript but we have specified this in the main text too. See Lines 157: Table S1.

3) The authors identified differences in the z-scores between phocids and otariids, with phocids showing higher integration than the latter. However, the type of data used to calculate such scores is not the same for both groups: in phocids, they used size residual scores, and in otariids, they used the superimposed coordinates. It is unclear whether the fissipeds used size residuals or superimposed coordinates since it is not mentioned. This is an important issue because using size residuals in one group but not in the other, regardless of the non-significant association between allometry and vertebral shape in otariids, may have inflated the integration results. The authors must ensure that both variables are comparable between groups and standardize the data to confirm the reliability of their findings.

We really apologize for not specifying this in the past version of the manuscript and the confusion that this has caused to the reviewer. All the analyses were computed with size residuals to assure comparability among groups because we had the same concern as the one experienced by the reviewer. We have clarified this across the results Lines 306-309. We have also specified that the Z-scores of fissipeds were also calculated from residuals (Lines 277-278).

Specific comments:

4) L 48-50. Include a citation for this sentence. Wellik & Capecchi (2003) are cited later at the end of the phrase, but their work did not provide a characterization of backbone regionalization in "land-going vertebrates".

corrected. Lines 50,52.

5) L 61: correct "has" to "have".

Changed. Line 61.

6) L 61. "That" needs a complement here.

Changed. Lines 61-62.

7) L 63-70. Simplify and split the sentence for clarity. Example: "The last two modules likely relate to locomotor performance, as they appear to be related to motion capability of the presacral spine 16. While the vertebrae of the anterodorsal module (...) from the 'Diaphragmatic joint complex' 17-20."

Splited, thank you. Lines 63-65.

8) L70-75. These sentence confusing, please clarify it.

Clarified. Lines 72-74.

9) L 80. "Effects of gravity" are not absent underwater. Instead, animals moving in this medium endure gravity and buoyancy. Rephrase for precision.

Changed. Line 76.

10) L 81-83. Clearly state that such a situation only applies while underwater since aquatic tetrapods can likewise spend time on land and support their weight/ move by generating friction with the substrate.

Changed. Lines 77.

11) L 83-85. Improve the sentence so to better link it to the previous information. Example: In the axial skeleton of some marine mammals, such as cetaceans, secondary aquatic adaptations involved the reduction of regionalization including (...).

According to the comments of the first reviewer, this sentence has changed. Lines 82-84.

12) L 76-79. This sentence here weakens the novelty of your work by stating that is a continuation of previous studies but on a different clade. I believe that it should be presented only after providing the state-of-the-art on the integration and modularity of the vertebral column in pinnipeds. By doing so, the introduction now highlights the innovation of addressing these questions in aquatic carnivores, making your biological question much more interesting. I recommend moving this sentence to a new last paragraph of your introduction, where your goals are clearly stated.

Ok. We have moved this sentence (Lines 115-119) and we have also included a paragraph stating the importance of studies on integration and modularity (Lines 119-128). We have also clarified that there are no studies on the integration and modularity of the vertebral column in pinnipeds and, therefore, the findings of this paper are entirely novel (Lines 110-114).

13) L 90-93. The use of “by integrating” here is odd. Please rephrase the sentence. One possible alternative: “demonstrate a reduction in the number of modules in which lumbar, sacral, and anterior caudal are integrated into a single (...)”.

Changed, thank you. Line 84.

14) L 95-97: Repetition of lines 86-89, exclude or restructure paragraphs.

Re-structured. Lines 141-154.

15) L 98: fix misspelling in ‘less’

Fixed, thank you. Lines 141.

16) L 97-100. This should also be better placed in a last paragraph of objectives and hypothesis.

Moved to the last paragraph, thank you. Lines 141-154.

17) L101-107. This paragraph is lost here. It’s better suited after L 89.

This paragraph has been re-structured Lines 89-109.

18) L 109: This count represents, in average, how many vertebrae per species? Also, precise how many species you used and the range of individuals/ sp.

This paragraph. Thank you. Lines 157-161.

19) L 152: Were all fissipeds extracted from Martin-Serra et al 2021? Precise.

Yes, and it has been precised in the text (Lines 206-207) and in the caption of Figure 3 (Line 276).

20) L 175-176: This premise came as a surprise, nowhere in the introduction you mention hypaxial and epaxial muscles. Make sure you present such assumptions in the introduction.

We have mentioned this across the introduction. See response to comment #2.

21) L 201-202: Where fissiped data comes from should be stated in the methods as well.

It is mentioned in Lines 206-207.

22) L 280 Figure 4. Remove “of”.

This has changed according to the comments of R1.

23) L 298-300 This affirmation needs to be tempered. Comparing these parameters to only one fissiped species is not enough for such generalization.

We have tempered this affirmation. See Lines 358-361.

24) L 304-305. Yes, but your parameters are not quite comparable. Se general comment 3.

See response to comment #3

25) L 305-307. Again, this affirmation needs to be tempered.

We have tempered this affirmation. See Lines 413-414.

26) L 340: Correct to “swimmers do not need to support their (...)”

Corrected. See Line 392.

27) In the results: integration results of fissipeds are not described (just presented in Figure 3). Did you use size residuals or Procrustes coordinates? Precise.

Precised. See Line 277-278.

28) Figure 1: What is A B and C? Precise which species, axial region, and anatomical orientation.

Precised. See new Lines 184-187, now is Figure 2.

29) Figure 3. Increase axis text, and describe what the abbreviations represent. B) correct “phocids” to phocids. C. Isn’t it “fissipeds” and “pinnipeds” instead of “phocids” and “otariids”? Use bars alongside the axes to indicate the region to which the vertebrae belong to.

Corrected.

REVIEWER #3 (comments in text highlighted in blue)

This is an elegant morphological data set comparing vertebral column integration and surrounding musculature in seals and otters, and within the seals comparing the true seals and sea lions / fur seals. This paper uses specimens from collections to quantify morphometrics along the vertebral column and CT scans of living specimens to quantify epaxial and hypaxial musculature among groups. The authors found that while seals are integrated, they are less integrated than otters. Within the seals, the true seals are more integrated than the sealions/ fur seals. The authors discuss the functional modules or lack of functional modules in the different groups and eventually bring these ideas back to address locomotion and evolutionary transitions.

Thanks for the comments regarding our MS and for the constructive review. Below, we answer all reviewer concerns point-by-point.

Major comments:

The morphology in this paper is really lovely, but this paper in its current form is very technically related to pinnipeds and fissipeds. I think the paper, the intro specifically, might need to be reframed to be more obviously addressing land to water evolutionary transitions and locomotion differences in these groups. I am not sure this current version would appeal to the broad readership of the selected journal. I know space is limited, but I think this paper would benefit from a schematic including a phylogeny of these groups showing how they are related and how their locomotion might differ. This would be much more meaningful to me than Tables 2 and 3, which show Z scores. I think these tables should be supplemental. Readers can glean these from the results section and figure 3.

We have reframed the introduction. Accordingly, we have specified more explicitly how worth is the study of the impact of the land-to-water transition on the evolutionary integration and modularity of the axial skeleton (See Lines 79-82). Moreover, we have also extended the differences in the locomotion between the extant groups of crown pinnipeds (Lines 89-109; the text is highlighted in green because this point was also raised by Reviewer 2).

We have also included a new Figure, Figure 1, showing of these groups showing how they are related and how their locomotion might differ.

Minor comments:

1; Line 29: define fissipeds

Defined. See Lines 29-30.

2; Line 63-70: This sentence is very long and difficult to follow. It might be best to break these ideas up for easier consumption by the reader.

Yes, this sentence was changed already according to comments of R1 and R2. See Lines 63-69.

3; Line 70 and 74: It would be useful to give an definition of the Diaphragmatic joint complex.

Defined. See Lines 69-70.

4; Line 81-83: For clarity, I suggest saying fully aquatic mammals (living entirely in water), which can contrast to partially aquatic mammals, or something similar, to refer to other species. As it is written, many of the animals talked about in the intro generate friction with the substrate when they are moving on land.

We agree with the reviewer in this point. We have included “while underwater” according to the comment of reviewer #2. See Line 77.

5; Line 86: maybe revise to say ‘pinnipeds, partially aquatic mammals,...’ These changes make evolutionary and function groupings clear to the readers of a large broad journal.

Yes, this has been clarified in the paragraph 89-109.

6; Line 98: ‘less’

Corrected. See Line 141.

7; Line 101: should this list include 'and'?

Yes. Thank you. This sentence has changed according to R1 and R2.

8; Line 151: replace 'ref' with author's names

Changed. See lines 202 and 208.

Results: It is confusing that Fig 3A and 3C are first compared. Consider re-lettering the figure so that you can present 3A and 3B first in the results section and then 3C and 3D.

Yes, we have changed this. Thank you. See new Figure 3.

9; Figure 4: This is really interesting and well presented.

Thanks.

10; Line 329: The hinge is a really cool idea! The following paragraphs are really interesting. This paper would benefit from moving some of these ideas to the intro as a functional hypothesis to be tested throughout this study.

Thank you very much for appreciating our work, but all these ideas came from the results obtained in this paper and they are presented as functional explanations to the patterns found. New hypotheses to test in the future. However, in our opinion moving these ideas to the introduction as a functional hypothesis to test would be an ad-hoc procedure. Therefore, we prefer to keep our original hypotheses in the introduction. It is true that our original hypothesis was diluted (not specified) across the introduction in the previous version of the ms) but in this new draft we have made an effort to expose them more clearly. Hope that now the reviewer would be happy with our original hypotheses.

REVIEWERS' COMMENTS:

Reviewer #1 (Remarks to the Author):

This is a review of the second version of this manuscripts. The authors employed 3D Geometric morphometrics to investigate the integration of vertebral shape along the column of Pinnipeds, and compare it to a previous work done on Fissipeds.

The authors have done a great job incorporating all the reviewers' comments into this new version. I have included minor comments along the pdf.

I still believe that coherence and cohesion among paragraphs could be improved. In some cases, both in the introduction and the discussion the authors do not manage to give the text a coherent flow. This prevents the reader from missing the main point or following the authors' thinking. See comments on pages 6 and 19, but see comments along both sections.

M&M section has been greatly improved. Despite this, it would also benefit from improvements directed towards improving clarity, coherence and flow (see comments on pdf file)

I understand that the authors are not native speakers of English, and being a non-native English speaker myself, I acknowledge their effort. It is extra work for us to be able to communicate our work clearly. Even though they make good use of most grammar structures, there is room for improvement in particular complex sentences (please see comments in the file). Moreover, I believe that the texts would greatly benefit by having shorter sentences as current sentences tend to extend too much and may me confusing in some cases. Try to separate sentences, I have made some edits on this regard. In addition to this, I have suggested variation in the use of connectors and conjunctions. In order to aid the authors in this endeavor, I have made suggestions regarding grammar all along the document, including some proposed rephrasing.

Overall, I believe that the authors have done a great first step towards improving the manuscript. Despite this, I believe that having these minor writing changes done would be necessary for it to be publishable.

Best regards

Reviewer #3 (Remarks to the Author):

I really enjoyed reading this paper the first and the authors have done a fantastic job with the revisions. I think the additional text broadens the scope of this paper. I think the revisions to the initial hypotheses in the intro make the goals of the paper more clear. I am still excited about the hinge hypothesis in the discussion. I look forward to seeing this in print. Congrats to the team!

REVIEWER #1 (comments in text highlighted in yellow)

This is a review of the second version of this manuscripts. The authors employed 3D Geometric morphometrics to investigate the integration of vertebral shape along the column of Pinnipeds, and compare it to a previous work done on Fissipeds. The authors have done a great job incorporating all the reviewers' comments into this new version. I have included minor comments along the pdf.

Thank you very much for your comments and for the correction you had made. We have incorporated all your minor comments into this final version. Here you will only see our response to the most relevant ones, but we have incorporated all suggestions into the manuscript (highlighted in yellow).

I still believe that coherence and cohesion among paragraphs could be improved. In some cases, both in the introduction and the discussion the authors do not manage to give the text a coherent flow. This prevents the reader from missing the main point or following the authors' thinking. See comments on pages 6 and 19, but see comments along both sections.

We included the reviewer suggestions to improve coherence and cohesion.

M&M section has been greatly improved. Despite this, it would also benefit from improvements directed towards improving clarity, coherence and flow (see comments on pdf file). I understand that the authors are not native speakers of English, and being a non-native English speaker myself, I acknowledge their effort. It is extra work for us to be able to communicate our work clearly. Even though they make good use of most grammar structures, there is room for improvement in particular complex sentences (please see comments in the file). Moreover, I believe that the texts would greatly benefit by having shorter sentences as current sentences tend to extend too much and may be confusing in some cases. Try to separate sentences, I have made some edits on this regard. In addition to this, I have suggested variation in the use of connectors and conjunctions. In order to aid the authors in this endeavor, I have made suggestions regarding grammar all along the document, including some proposed rephrasing. Overall, I believe that the authors have done a great first step towards improving the manuscript. Despite this, I believe that having these minor writing changes done would be necessary for it to be publishable.

We really appreciate the guidance. We have incorporated all grammatical corrections recommended by the reviewer.

What do you mean by that? in which sense the regional proportions are "conserved"... This part is not clear.

We meant regional identities; we have changed this sentence to improve clarity (Line 86).

In this grammar structure there is no need to include the subject again.

Changed. (Line 88)

"In this sense"? this is used two paragraphs above.

True, we changed the connector. (Line 89)

This shouldn't be a new paragraph. Even though this is not indented is clearly disconnected from the previous sentence, despite the link between the two.

OK, we joined it with the upper paragraph. However, we believe that there is a connection because we are discussing the movement of phocids and in the subsequent sentence, we mention the features of the spinal column that play a role in their locomotion. (Line 94- 103)

Either you add "the fact that" or you revise the sentence for grammatical correctness. The way it is right now is not grammatically correct... You could also change "despite" for "even though".

OK, changed to “even though”. (Line 104)

One sentence = one paragraph? See comments below

Yes, we merged it with the subsequent paragraph. (Line 104-108)

I would move all this to the previous paragraph, leaving a paragraph explaining modularity, integration and their importance and another paragraph explaining your goal and what you did.

True, we moved it. (Lines 108-118)

Explain here why the direction of the fibers can be related to integration.... so when we get to the discussion say that "more disparate distribution of bundles suggest a less integrated region"...why? what is the reasoning behind this? You should mention it here or in the discussion.

We incorporated a sentence to clarify this reasoning. We would like to highlight that the reviewer mentions “direction of the fibers” but we only talk about “direction of muscle bundles”, which does not necessarily correlate with direction of fibers. (Lines 132-136)

I would move this phrase to the beginning of the next sentence before "Proportional". Here your hypothesis is based on the fact that both groups have distinct locomotor modes, not on the fact that the musculoskeletal system is key.... this is more related with the following sentence.

Moved (Lines 145-146)

This doesn't make sense...if you want to compare Fissipeds vs pinnipeds...you already made your predictions.... and I do not see the need to make a prediction about otarids and phocids...since your expectations are already stated in your hypothesis.

OK, removed.

I would move this on paragraph below...first explain the three different computations

Done (Lines 171-174)

Here, include the paragraph about geomorph and the phylogeny.

Done (Lines 171-174)

After performing a Procrustes superimposition, what you have are procrustes coordinates... be careful because you are using the term "x, y, z coordinates" to refer both to raw data and to Procrustes coordinates...

Yes, of course, it was a mistake. Corrected. (Line 177)

To compare the integration strength (Z-score) obtained for pinnipeds in this study with that obtained by Martín-Serra et al 16 for fissipeds, we repeated the analyses of integration after removing the landmarks digitized on the transverse processes of lumbar vertebrae (L16, L17, L33, L34) and on the ventral part of the centra (L35-L40)

Changed according to reviewer suggestion. Thanks. (Lines 187-191)

Why is this a different paragraph? And why is it in Present?

We merged with the subsequent paragraph and change into the past tense. (Line 192-206)

Again, try to avoid citing the same package over and over....if you include it at the beginning you could save a lot of text.

OK. Checked and corrected.

I believe these two verbs should be in the past. (Line 214; Line 216)

Yes, we change the whole paragraph into the past tense. (Line 192-206)

This is Mat & Met.... you should state your purpose..but what you did: "We assessed the existence of differences.....". I would remove the word qualitatively...since, even though it's just a ratio and is not highly precise...you are quantifying something.... not just looking at it and saying if it's larger in one or the other.

We eliminated the word qualitatively. (Line 214)

Is there a guideline from the journal saying that the Result section should be in the "present"? If not, please write in past, this is a work that you already did.... your results showed... in your dataset the allometric effect was significant.... Maybe it's just me...but I don't feel comfortable reading results in Present tense, If this is accepted by the editor, please ignore my comment.

Thanks for this, we are not aware of any journal guidance specifying the need for present tense, so we have chosen to follow your advice and shift it to the past tense.

Even though there was no apparent pattern in the non-phylogenetic analyses, it arises...

Changed according to reviewer suggestion (Line 226-227)

I do not believe this is well used, you are not adding contrast.... you're adding more information. They have an integrated thoracic region and this region includes an integrated diaphragmatic vertebrae

True, changed by "In addition". (Line 313)

You set the basis for this hypothesis in the previous paragraph...why is this a new paragraph? You take about the diaphragmatic vertebra two paragraphs above. Is there a way you could reorganize these three later paragraphs so you are not jumping between

paragraphs to explain one idea. Creating ONE paragraph to focus specially on the diafragmatic vetebra both in fissipeds and pinnipeds would be a good way to start. you talk abot this two paragraphs above...try to merge these paragraphs, so the inforation doesn't get repeated and you don't come and go from one paragraph to the other.

Yes,, we merged those paragraphs into one as suggested (Lines 306-346).

I'm still struggling to follow the reasoning behind this statement.

We incorporated a sentence to clarify this statement in lines 352-354.

Keep referring to the Vertebral column as "Vertebral Column"

Done.

REVIEWER #3

I really enjoyed reading this paper the first and the authors have done a fantastic job with the revisions. I think the additional text broadens the scope of this paper. I think the revisions to the initial hypotheses in the intro make the goals of the paper more clear. I am still excited about the hinge hypothesis in the discussion. I look forward to seeing this in print. Congrats to the team!

**Thank you very much for this comment and for the useful corrections you made.
We have done our best to improve our manuscript.**